# Micro-osteoperforation for enhancement of orthodontic movement: A mechanical analysis using the finite element method

João Ricardo Cancian Lagomarcino Gomes[1]*, Ivana Ardenghi Vargas[2], Antônio Flávio Aires Rodrigues[1], Luiz Carlos Gertz[3], Maria Perpétua Freitas[1], Sergio Augusto Quevedo Miguens, Jr.[1], Ahmet Ozkomur[1], Pedro Antonio González Hernandez[1]

1 Graduate Program in Dentistry, Universidade Luterana do Brasil (ULBRA), Canoas, RS, Brazil, 2 School of Dentistry, Universidade Luterana do Brasil (ULBRA), Canoas, RS, Brazil, 3 School of Mechanical Engineering, Universidade Luterana do Brasil (ULBRA), Canoas, RS, Brazil

* jr.cancian@hotmail.com

## Abstract

### Background

Micro-osteoperforation is a minimally invasive technique aimed at accelerating tooth movement. The goal of this novel experimental study was to assess tooth movement and stress distribution produced by the force of orthodontic movement on the tooth structure, periodontal ligament, and maxillary bone structure, with and without micro-osteoperforation, using the finite element method.

### Materials and methods

Cone-beam computed tomography was used to obtain a virtual model of the maxilla and simulate the extraction of right and left first premolars. Three micro-osteoperforations (1.5 x 5 mm) were made in the hemiarch on the distal and mesial surfaces of upper canines, according to the power tip geometry of the Propel device (Propel Orthodontics, Ossining, New York, USA). An isotropic model of the maxilla was fabricated according to the finite element method by insertion of mechanical properties of the tooth structures, with orthodontic force (1.5 N) simulation in the distal movement on the upper canine of a hemiarch.

### Results

Initial movement was larger when micro-osteoperforations were performed on the dental crown (24%) and on the periodontal ligament (29%). In addition, stress distribution was higher on the bone structure (31%) when micro-osteoperforations were used.

### Conclusions

Micro-osteoperforations considerably increased the movement of both the dental crown and periodontal ligament, which highlights their importance in the improvement of orthodontic movement, as well as in stress distribution across the bone structure. Important stress absorption regions were identified within micro-osteoperforations.

**Data Availability Statement:** All relevant data are within the manuscript and its Supporting Information files.

**Funding:** The author(s) received no specific funding for this work.

**Competing interests:** The authors have declared that no competing interests exist.

## Introduction

Research into orthodontics has been conducted to find materials, methods, and techniques that aid in tooth movement and shorten orthodontic treatment length, without affecting the tooth supporting structures, maintaining the integrity of the tooth root, periodontal ligament, and alveolar bone.

The techniques employed to accelerate orthodontic treatment were categorized into surgical and non-surgical. Non-surgical methods include self-ligating braces, single-arch aligners, and intraoral microvibrators. Surgical methods, on the other hand, consist of corticotomies, piezocisions, and micro-osteoperforations (MOPs).

The surgical method for accelerating orthodontic tooth movement is based on the regional acceleratory phenomenon (RAP) described by Harold Frost [1] as a tissue reaction to any noxious stimulus that increases healing. The basic assumption about surgically assisted orthodontic tooth movement acceleration is to induce trauma to the bone in the region where acceleration is needed.

MOP is a minimally invasive technique that does not require incisions and flaps, unlike conventional corticotomies or piezocision, and it can be performed by the orthodontist, thus considerably reducing treatment costs and providing a more comfortable and less painful postoperative period.

Numerous studies published in the scientific literature have investigated biological mechanisms of MOPs; however, there are no reports about the mechanical behavior of this type of intervention, especially about the finite element method (FEM).

The aim of the present study was to investigate initial tooth movement and stress distribution produced by orthodontic movement forces applied to the tooth structure, periodontal ligament, and bone structure, with and without MOPs for acceleration of tooth movement. The investigation was made by FEM.

## Materials and methods

A cone beam computed tomography (CBCT) image of a patient from a center for diagnosis and imaging was used for this laboratory experimental study. The research project of the laboratory experimental study conducted in partnership with the Graduate Program in Dentistry and the School of Mechanical Engineering of the Universidade Luterana do Brasil (ULBRA/Canoas–RS) was approved by the Research Ethics Committee (CEP-ULBRA) on February 20th, 2020, process CAAE 26665419.8.00005349.

The Center for Diagnosis and Imaging (CDI) authorized the use of cone beam computed tomography (CBCT) images, saved as STL files. From March 2020, the CBCT images were selected from the image bank of the clinic by the technical supervisor. The STL file generated the virtual models of the maxilla and mandible for the study entitled MICRO-OSTEOPERFORATION FOR ENHANCEMENT OF ORTHODONTIC MOVEMENT: A MECHANICAL ANALYSIS USING THE FINITE ELEMENT METHOD. The researchers were blinded to the CBCT images and to any other clinical and personal data of the patients from whom the CBCT images were obtained.

The image was selected according to the following eligibility criteria: patient with all 28 permanent teeth, bone and tooth anatomy compatible with normal patterns, absence of pathologies, and no history of facial trauma and/or previous orthodontic or surgical treatment.

After selection of the CBCT image from an ANONYMOUS DICOM file using MIMICS and SLYCER 3D software programs, a virtual model of the maxilla was built, establishing the 3D anatomical contours for the teeth, periodontal ligament, and cortical and medullary bones in STL files.

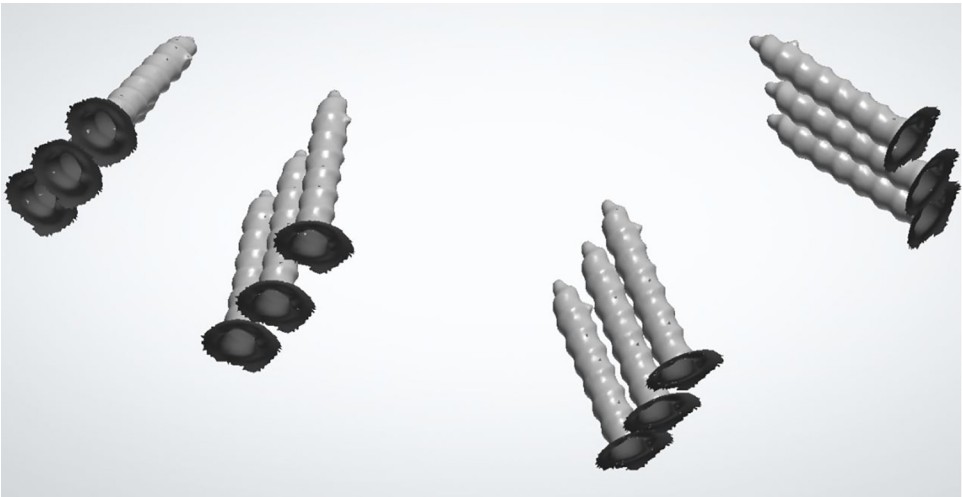

**Fig 1. PROPEL's power tip scanning.**

To perform MOPs, the power tip geometry and shape of the Excellerator RT Tip (Propel–Propel Orthodontics LLC, Ossining, NY, USA) were obtained by a bench scanner (SIRONA, model InEos X5) and later imported to the BLUE SKY PLAN software program, positioned on the virtual maxilla at predefined sites (Fig 1).

After that, the file containing the maxilla with the positioned perforations was exported to MESHMIXER for a negative Boolean operation and consequent simulation of perforations, extractions, and filling of the alveoli (Fig 2).

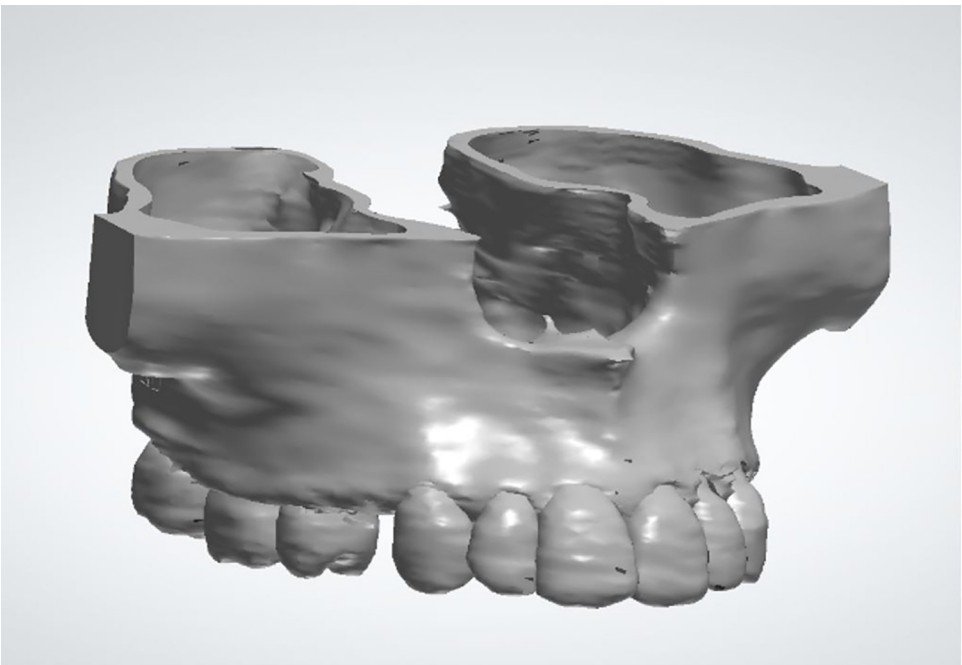

**Fig 2. Virtual model of the maxilla without MOPs.**

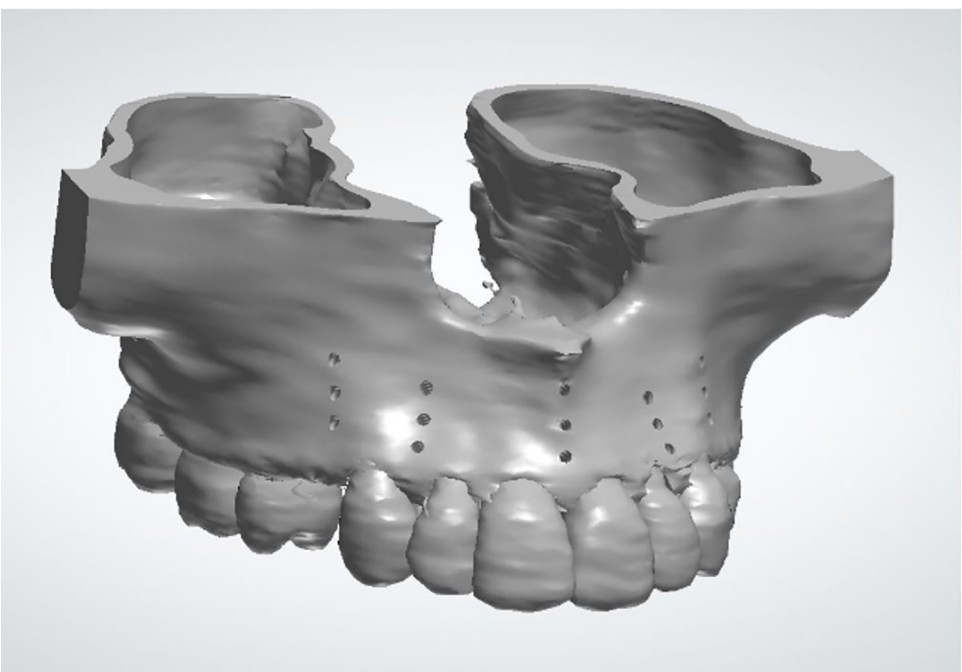

**Fig 3. Virtual model of the maxilla with MOPs.**

Extraction of first premolars was simulated in the maxillary hemiarch and three MOPs were performed, with linear distribution on the distal and mesial surfaces of the canines, as suggested by Blasi and Pavlin [2]. Each perforation had 1.5 mm in width and 5 mm of bone depth, simulating the power tip shape of the PROPEL device. On average, the distance between the perforation location and the periodontal ligament was 1 mm, and the spacing of the perforations was 4.5 mm (Figs 3 and 4).

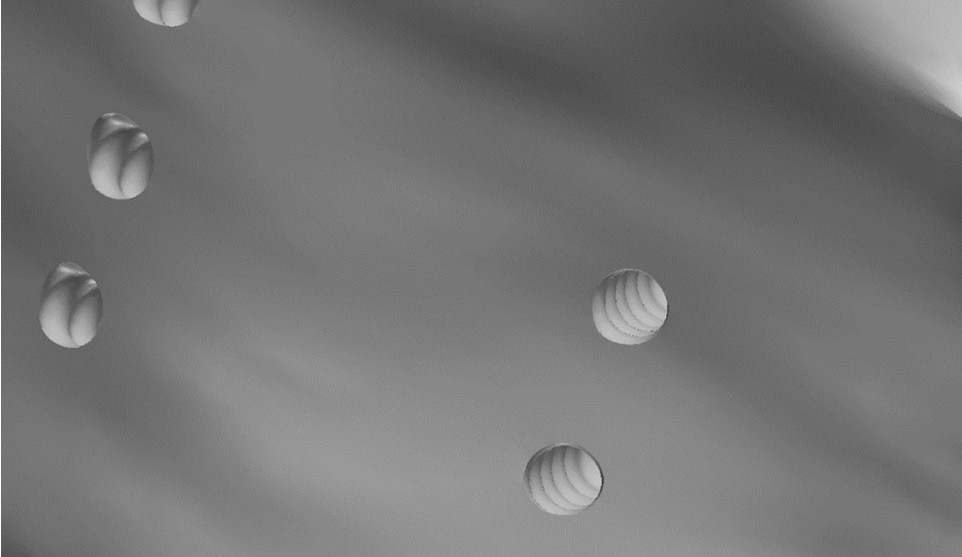

**Fig 4. Close-up view of the MOPs with the turns produced internally by the power tip of the PROPEL system.**

**Table 1. Poisson coefficient, density, young modulus of teeth, periodontal ligament, medullary and cortical bones, and granulomatous tissue.**

| Variable | Poisson coefficient | Young modulus | Reference |
|---|---|---|---|
| **Tooth** | 0.31 | 14,700 MPa | Tsoukinas, 2020 [4] |
| **Periodontal ligament** | 0.45 | 0.69 MPa | Serpe, 2015 [5] |
| **Medullary bone** | 0.3 | 1,370 MPa | Silva, 2010 [6] |
| **Cortical bone** | 0.3 | 13,700 MPa | Silva, 2010 [6] |
| **Granulomatous tissue** | 0.167 | 1 MPa | Isaksson, 2009 [7] |

From April to August 2020, there was an imposed pause in research activities due to COVID-19 restrictions, which were resumed from August 2020, with the STL file was imported to the SOLIDWORKS program, replicating the geometry of the 3D virtual model for obtaining an isotropic model, which included the Poisson coefficient and Young modulus of the investigated structures (Table 1), necessary for the FEM, as described by Rubin et al. [3].

Linear behavior and physical properties of the granulomatous tissue at the perforation sites were considered, assuming that perforated areas are filled by inflammatory phase elements (synthesis), advancing into the proliferative phase (differentiation) and later tissue maintenance (remodeling) [8] after surgically induced bone trauma.

Thereafter, two isotropic virtual models were built, one with MOPs and one without MOPs, obtained from the same model of a maxillary hemiarch, thus allowing for comparisons and a more appropriate control because there would be no anatomic differences between the models.

Using the ANSYS R19 software, a model mesh was placed on the isotropic model of the maxillary hemiarch, with tetrahedral elements interconnected by nodal points on an X, Y, and Z coordinate system, allowing orthodontic forces to be applied in the software that simulated tooth movement and the consequent stress distribution on the tooth and supporting structures (Fig 5).

Possible numerical results were assessed with and without the turns produced internally by the power tip in the MOPs, and no changes were found in stress distribution and tooth movement.

The Excellerator RT power tip was then streamlined and shaped into a cylindrical form, thus simplifying mesh creation and local refinements and allowing for an adequate amount of elements for the assessment of gradients across the segments, as suggested by Duprez et al. [9].

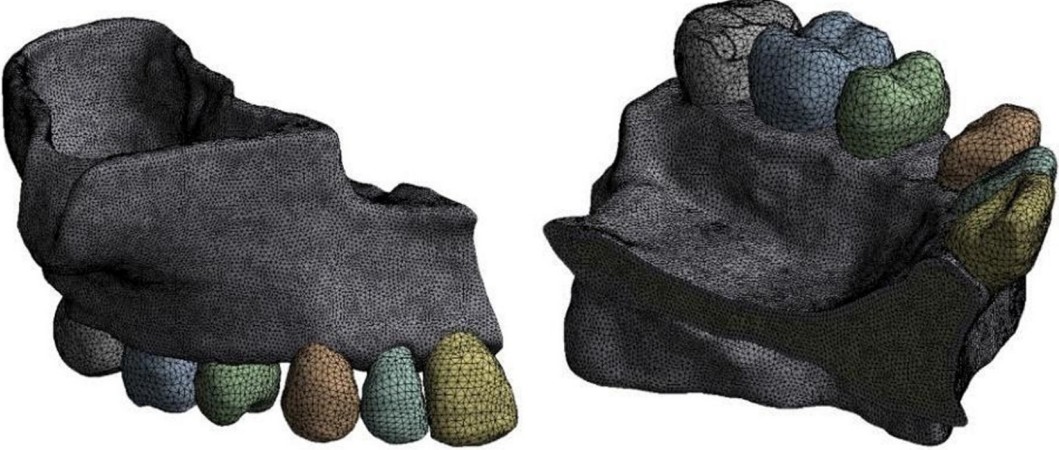

**Fig 5. Virtual image of the model mesh on the isotropic model of the maxillary hemiarch without MOPs.**

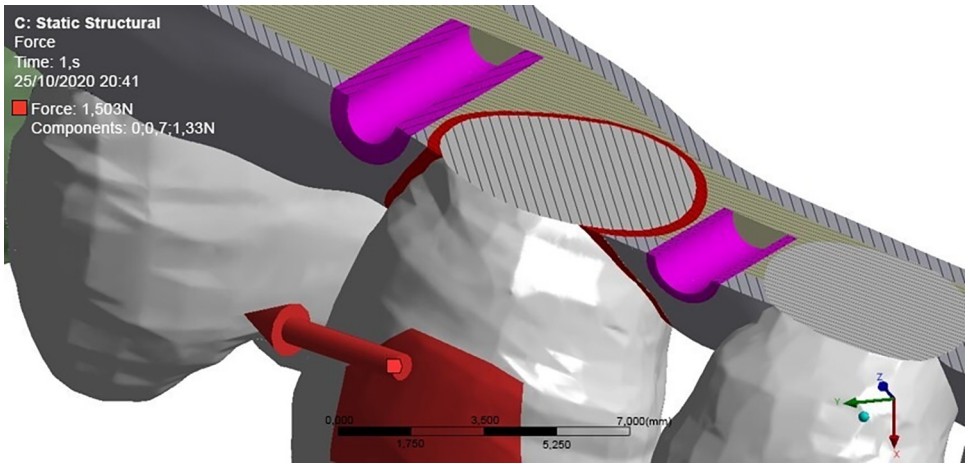

**Fig 6. Image of the region showing the direction of the orthodontic retraction force.**

The effect of retraction force (1.5 N) (mesial-distal direction) simulated on the upper canine of a maxillary hemiarch (Fig 6) was assessed using the ANSYS R19 software, as recommended by Ren et al. [10] (Fig 6).

To reduce the tendency towards rotation in the buccolingual direction and distal inclination of the tooth to be distally moved, moments of opposite forces were applied to minimize undesired side effects and allow for a pure distal movement (Fig 7).

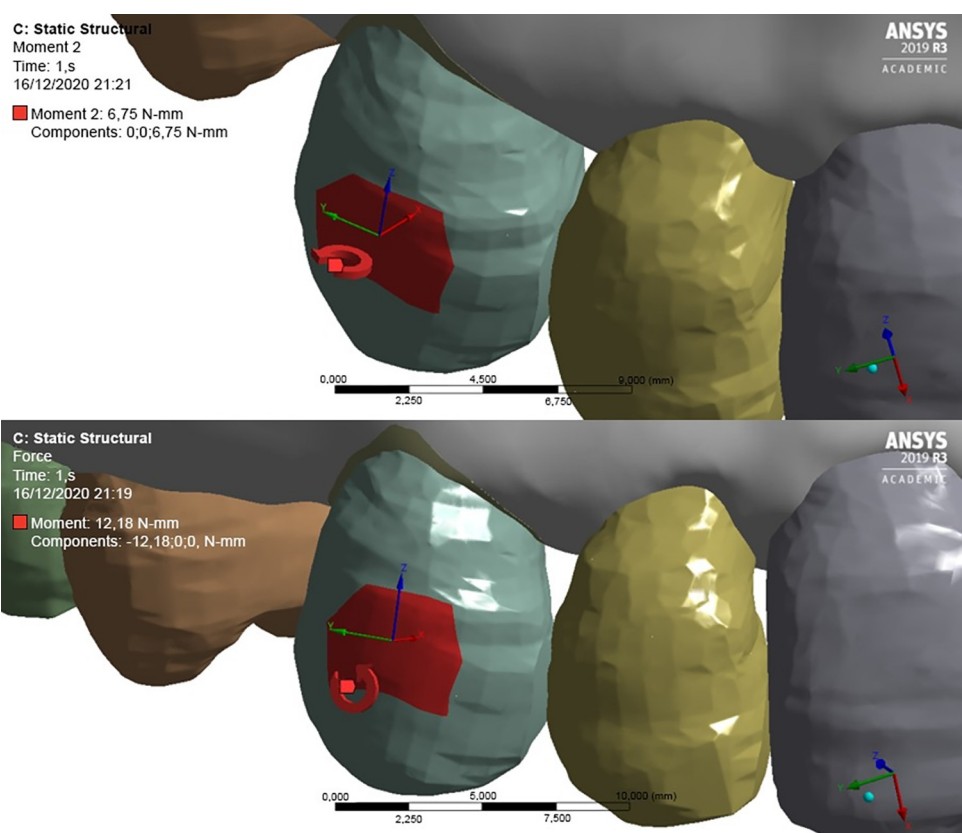

**Fig 7. Application of moments of counter-rotational force.**

Stress was analyzed using numerical modeling via FEM. The Von Misses stress criterion was used to assess stress distribution on the teeth, periodontal ligament, medullary and cortical bones, and granulomatous tissue, as described by Lotti et al. [11].

As pointed out by Blatt et al. [12] and Provatidis [13], the initial tooth movement (in mm) and stress distribution on the tooth and supporting structures (in MPa) were quantified and analyzed by graphical tables with colors created by the FEM software, where blue and red indicated the weakest and strongest intensity, respectively. Information on stress distributions allowed assessing the degree of strength to which each element was subjected. Given the methodology used in laboratory experimental studies, it was not possible to perform statistical calculations.

Once the laboratory phase of the research had been completed, in March 2021, the final review, analysis of the data, and writing of the manuscript were conducted, followed by the defense of the master's thesis in dentistry. All images that were produced but not selected for use in this article are provided as Supporting information (S1–S8 Files, S1–S6 Figs).

## Results

By analyzing the displacement produced immediately after application of the distalization force, the maximum value was 0.0686 mm without MOPs and 0.0900 mm with MOPs in the dental crown region, indicating a 24% higher immediate displacement with MOPs (Fig 8A and 8B).

The periodontal ligament showed a similar pattern, with a maximum displacement equal to 0.0455 mm without MOPs and 0.0645 mm with MOPs, but 29% higher with MOPs (Fig 9A and 9B).

Regarding maximum stress, its location, distribution, and concentration were different for the assessed structures. In the dental crown region, immediately after the application of the distalization force, there were regions of higher stress magnitude and concentration in the orthodontic bracket area, with no significant numerical difference either with or without MOPs. In the root region, there were small striated regions of slight stress along the root length, also without significant numerical difference for the presence of MOPs (Fig 10A and 10B).

Regarding stress intensity and distribution on the periodontal ligament, the stress magnitude pattern (light blue) observed in the cervical region decreased in intensity in the apical region (darker blue), regardless of whether MOPs were performed or not, with no significant numerical difference either with or without MOPs (Fig 11A and 11B).

In the alveolar bone, stress was distributed obliquely in the region of the tooth that received orthodontic distalization force, with higher stress (in orange) in the cervical/distal region and at the medial/mesial third of the tooth; zones that indicate areas of pressure, which represent cellular response to bone resorption (Fig 12A and 12B).

This oblique stress distribution was also observed on the cortical bone, and it was higher in the apical/mesial region. This demonstrates a tendency towards pendular motion after force application. However, stress distribution is higher with MOPs than without them. We observed areas of high stress distribution (in orange) on the mesial and distal surfaces of the canine tooth with MOPs, while without MOPs, we observed areas of intermediate stress distribution (in green) in the mesial region of the canine tooth (Fig 13A and 13B).

Some areas of high stress distribution were found on the distal surface of the alveolar bone, relative to the lateral incisor. This suggests MOPs increased the stress distribution produced by the distalization force on the bone structure, even in the opposite direction of the force (Fig 13B).

Areas of high stress distribution were also observed inside MOPs, on the cortical and medullary bones, because with MOPs the area of the cortical and medullary bone structure is

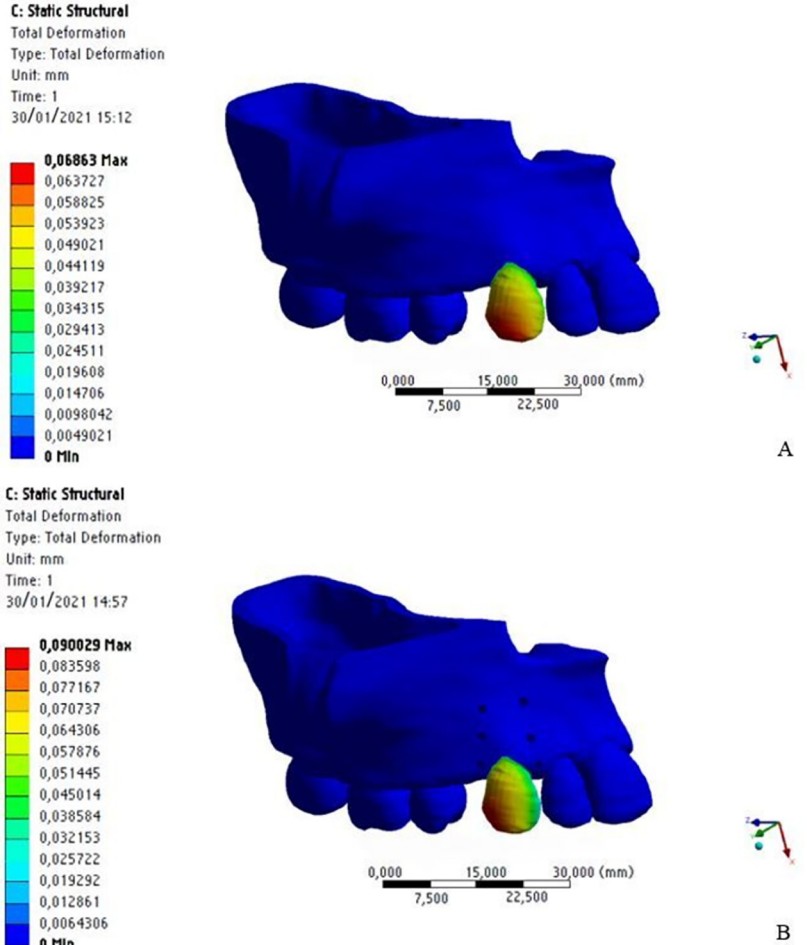

**Fig 8.** Displacements of the dental crown without MOPs (a) and with MOPs (b).

reduced. As stress represents force divided by area (S = F / A), consequently, stress is higher at the site of MOPs, when compared to the same site without perforations (Fig 14A and 14B).

Stress distribution on the medullary bone showed a similar behavior to that on the cortical bone, with areas of higher stress at the middle and apical/mesial thirds, indicating a tendency towards a pendular motion after application of orthodontic force. With MOPs, there was a considerable increase in stress distribution, with areas of intermediate stress (in green) distally to the canine tooth (Fig 14B).

With respect to the stress produced on the bone structure after application of orthodontic distalization force, a maximum stress of 0.25 MPa was observed without MOPs and 0.36 MPa with MOPs, with stress on the bone structure being 31% higher with MOPs (Figs 12B, 13B and 14B).

The values for immediate displacement on the dental crown and periodontal ligament with and without MOPs are illustrated as a comparative graph (Fig 15).

The values for maximum stress on the dental crown, root, periodontal ligament, and bone structure with and without MOPs are illustrated as a comparative graph (Fig 16).

## Discussion

To date, there are no published studies evaluating the mechanical components (stress intensity and distribution on the tooth and supporting tissues) produced during the acceleration of

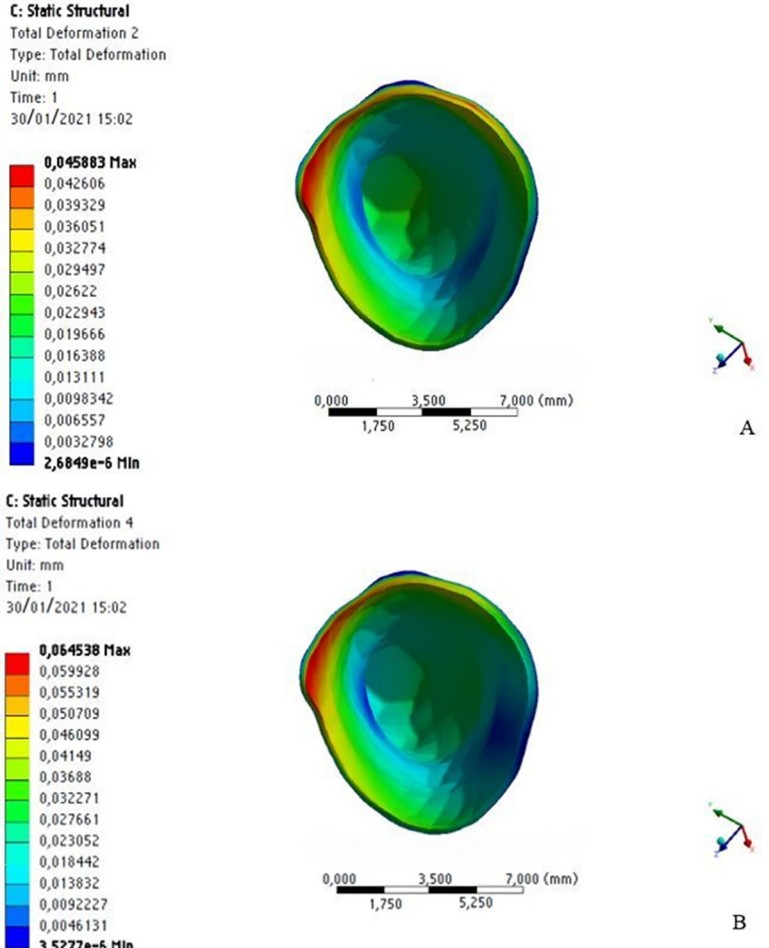

**Fig 9.** Displacements in the periodontal ligament without MOPs (a) and with MOPs (b).

orthodontic tooth movement using MOPs. Nevertheless, knowledge of mechanical and biological variables associated with orthodontic movement allows better understanding of tissue reactions during bone remodeling that help mitigate unrepairable and/or irreversible damage to tooth structures and to the periodontium.

As pointed out by Alikhani et al. [14], MOPs can increase tooth movement rate by 2.3 times, followed by significant increase in the levels of proinflammatory markers (chemokines CCL-2, CCL-3, CCL-5, and IL-8, and cytokines IL-1, TNF-α, and IL-6). They suggested higher levels of these cytokines are associated with major osteoclast activity and, therefore, larger tooth movement [15–17].

In another study, however, Alikhani [18] found no direct correlation between the increase in inflammatory exudate and larger tooth movement, because other factors also influence the amount of movement, such as root shape, alveolar density, occlusal forces, or design limitations of orthodontic appliances. However, all these variables are very hard to control in human studies because of their high variability, requiring large samples of individuals with the same anatomic characteristics, age, sex, and type of malocclusion, and these limitations may lead to erroneous conclusions in clinical trials.

The magnitude of the force used in this study to simulate the orthodontic distalization force (150 gf ≈ 1.5 N) has been very well described in the literature in numerous clinical trials, and

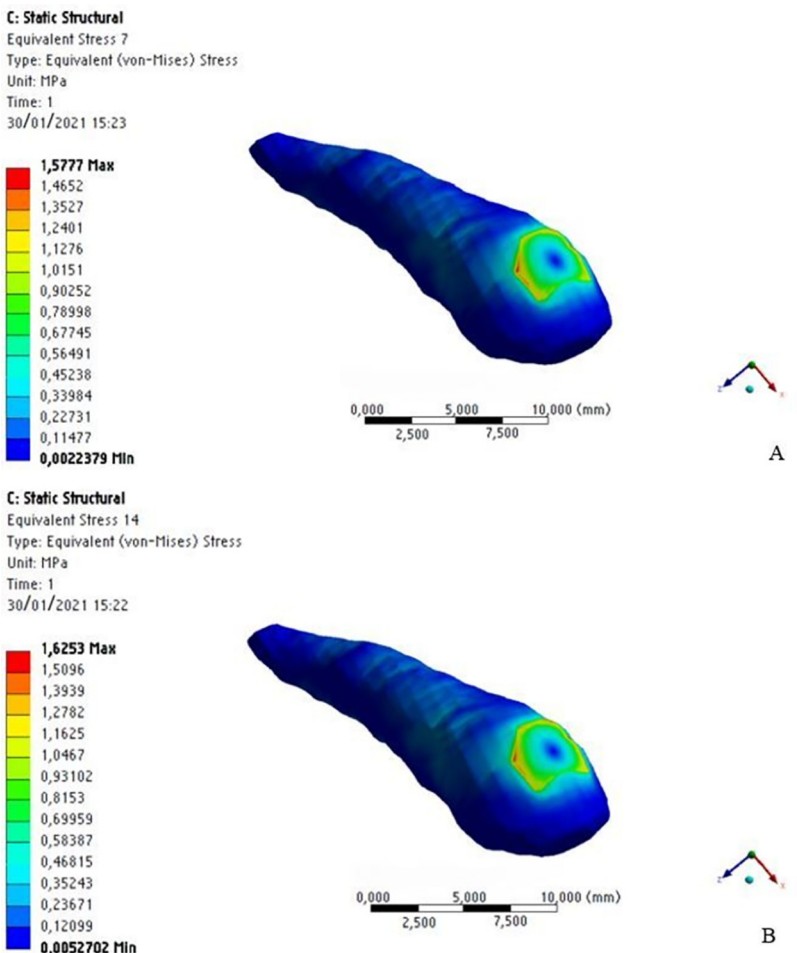

**Fig 10.** Stress distribution on the dental crown and tooth root without MOPs (a) and with MOPs (b).

it was not the objective of this study to discuss it or establish another degree of intensity to be used. We chose to use the Von Misses stress criterion to obtain a clearer result regarding the magnitude of maximum stress, as well as the stress distribution resulting from the simulated orthodontic force on the tooth and supporting structures.

The periodontal ligament was considered solid based on the methodology used in this study. According to Serpe et al. [5], the model used for the periodontal ligament (modulus of elasticity and Poisson coefficient) does not characterize the periodontal ligament as fluid. According to McCormack et al. [19, 20], periodontal ligament fibers play a key role in load transfer between the tooth and alveolar bone and their inclusion makes a significant difference to both the magnitude and distribution of strains produced in the surrounding bone; thus, they should be considered in FEM studies investigating the biomechanics of orthodontic tooth movement.

Accuracy in the construction of the virtual model was noteworthy, especially with regard to the characteristics of each structure involved, such as the teeth, periodontal ligament, and cortical and medullary bones. However, it is important to highlight that the technique developed by Propel was designed to be performed by orthodontists themselves, in an easier way, without incisions or flaps. The technique does not require the use of a rigid surgical guide, produced

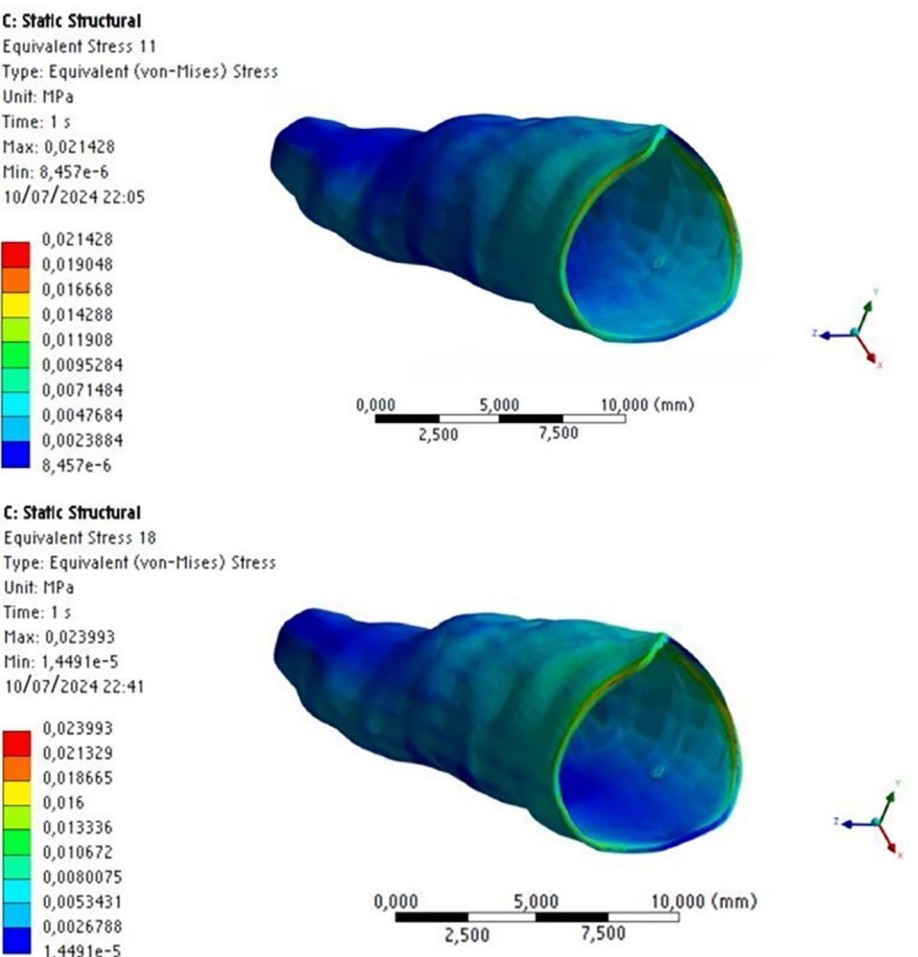

**Fig 11.** Stress distribution on the periodontal ligament without MOPs (a) and with MOPs (b).

by computed tomography and digital planning, to obtain more accurate drilling. The technique suggests making three linear perforations; however, performing MOPs may not be possible due to the convergence or proximity of the roots. Therefore, standardizing a minimum spacing of the perforations or a minimum distance between the perforation location and the periodontal ligament would not reflect the clinical reality of the technique.

Considering the paucity of reports in the literature on the mechanical behavior of MOPs, the present findings showed that removal of the bone structure by performing MOPs resulted in a bone that was more prone to deformation and movement with the application of orthodontic force, thus allowing increased tooth movement immediately after application of simulated orthodontic force, both on the crown and on the periodontal ligament, indicating greater MOP efficiency in the use and transformation of orthodontic force as well as in the displacement of the periodontal ligament to accelerate orthodontic movement, which, when related to biological principles, would result in greater bone remodeling.

The results of this study showed no difference in stress intensity and distribution in the roots or periodontal ligament either with or without MOPs. Due to the high-fidelity virtual model produced, stress was distributed on the bone structure through the periodontal ligament. Therefore, the study showed the conductive function of the periodontal ligament for

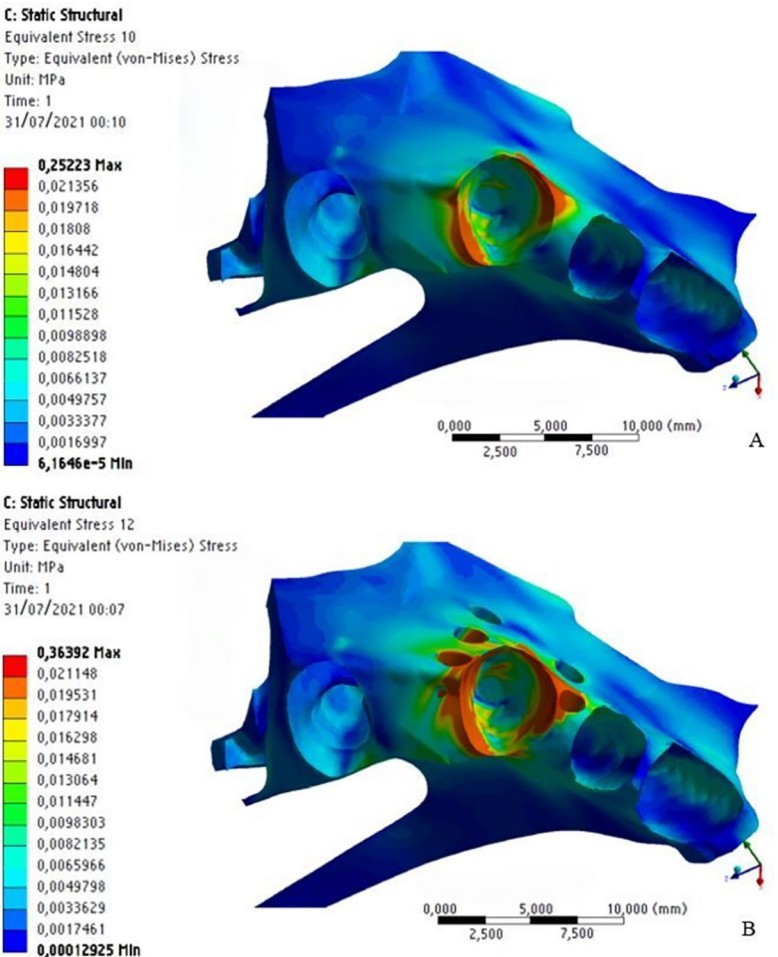

**Fig 12.** Stress distribution on the alveolar and cortical bones without MOPs (a) and with MOPs (b).

stress distribution produced by simulated orthodontic force, without the presence of areas of significantly high stress distribution on the root or periodontal ligament.

MOPs also led to higher stress distribution produced by orthodontic movement on the alveolar bone, which could indirectly protect the root structure since, when transferred to the biological model, this result could prevent or reduce inflammatory resorption. According to studies on root resorption due to induced tooth movement, the absolute magnitude of the applied force is not what matters the most, but rather the distribution of this force along the tooth root and adjacent bone structure. Force distribution by area of periodontal ligament is influenced by the type of movement performed, by the alveolar ridge morphology, and by root morphology [21–23]. Hence, perforations relieve the stress on the root structure and periodontal ligament and, consequently, allow for larger stress distribution and absorption. For Deng et al. [24], MOPs can protect the root surface if the observation time is extended until the end of tooth retraction.

Histologically speaking, the time necessary for cellular mobilization and the intensity of the mechanical stimulus to be translated by periodontal tissues should be analyzed. In the clinical interpretation of the results obtained in the present study, the time variable is crucial for cellular modification and differentiation, as well as for stress distribution by the periodontal

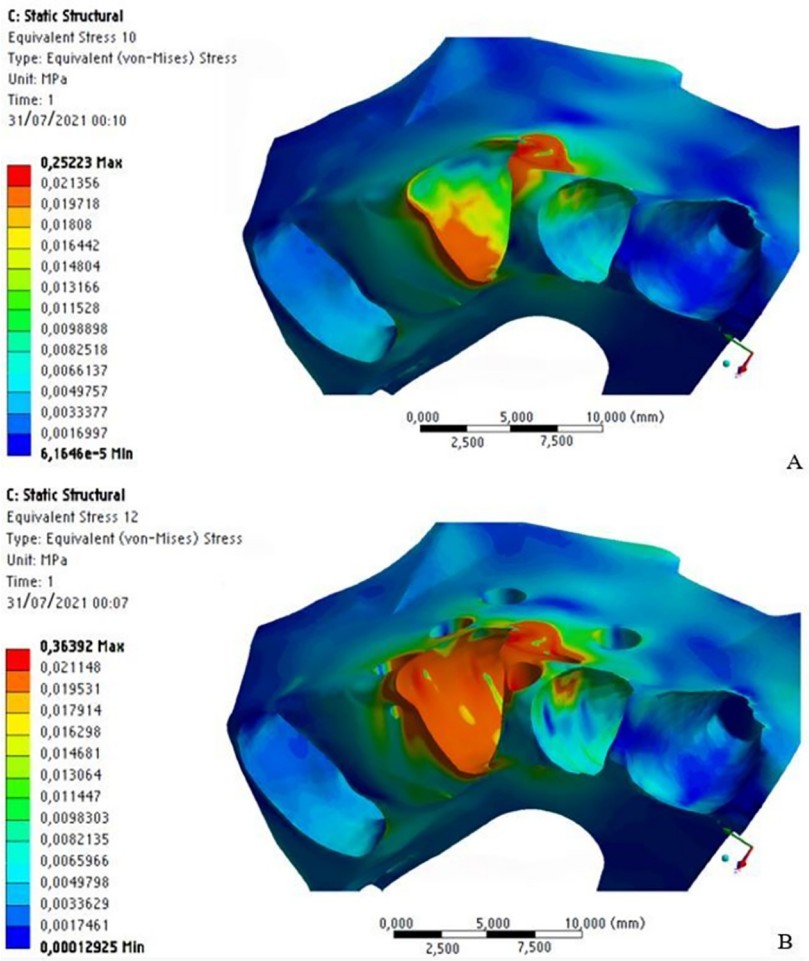

**Fig 13.** Stress distribution on the alveolar and cortical bones without (a) and with (b) MOPs.

ligament and alveolar bone. This phenomenon was reported by Krishnan and Davidovich [25] as "mechanotransduction," i.e., the capacity of tissues to respond biologically to a mechanical stimulus.

Moreover, it is widely known that tooth movement occurs through a controlled inflammatory process, triggered by the application of a certain force. MOPs can absorb the stress produced by orthodontic force as "escape zones," and also have lower resistance to the bone structure owing to the reduction of the area to be resorbed. Also, the great advantage of MOP is to provide a larger distribution of stress produced by orthodontic movement on the alveolar bone.

Given the difficulty in conducting clinical trials because of the complexity to standardize a sample and also to quantify and qualify the results, Santos et al. [26] in their meta-analysis, reported the sample sizes in studies on MOPs show great variability and could cause differences between the experimental groups, especially regarding the differences in the frequency of performing MOPs, follow-up periods, perforation systems, and types of retraction, which statistically demonstrated large clinical and methodological heterogeneity of the studies included in their meta-analysis.

Study designs that use the Propel system with appropriate sample sizes and that assess the effects of MOPs throughout total anterior retraction can generate results with better evidence.

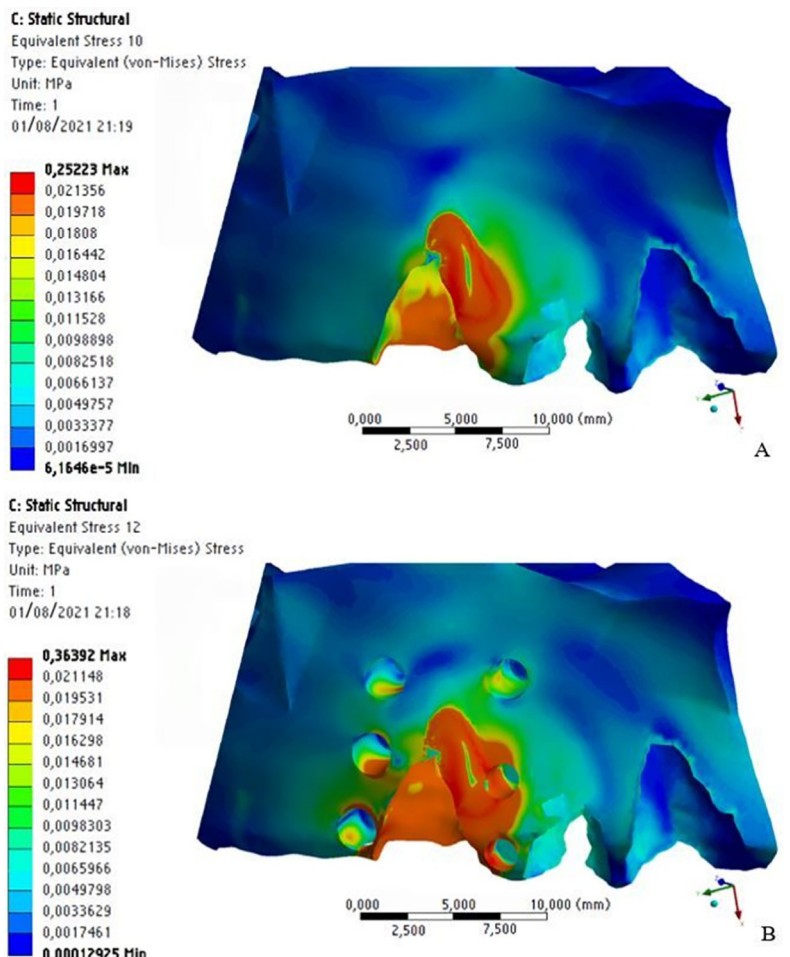

**Fig 14.** Stress distribution on the alveolar and medullary bones without MOPs (a) and with MOPs (b).

Thus, the present study on the use of the Propel system allowed replicating the anatomical structures and MOPs, as well as the orthodontic force for simulation of tooth movement, taking into consideration knowledge of physiological processes of repair and tooth movement in order to allow understanding the findings, in line with Middleton et al. [27], who claimed that quantitative data on initial movement of the tooth element and on stress distribution around it can be accurately predicted and be used to assess orthodontic treatment outcomes.

Considering the use of the FEM, the major limitation of this study was the inability to reproduce cellular and tissue responses using a mathematical model due to the specificities and complexity of oral and craniofacial structures. In this respect, this study seems to fill a gap of previously published clinical trials in terms of physical and mechanical evidence on enhanced orthodontic movement produced by MOPs, as described by Attri et al. [28]. MOPs demonstrated effectiveness in accelerating orthodontic tooth movement by initially increasing tooth displacement not only by the applied forces and cell activation and migration, as shown in clinical and biological studies, but also by the subtraction of bone structure, producing less dense bone tissue. Extrapolating to clinical practice, this will likely result in a joint contribution of mechanical and biological factors to accelerate bone remodeling and, consequently, tooth movement. Further studies on FEM and clinical evaluation are needed to elucidate all mechanisms and clinical outcomes of MOPs.

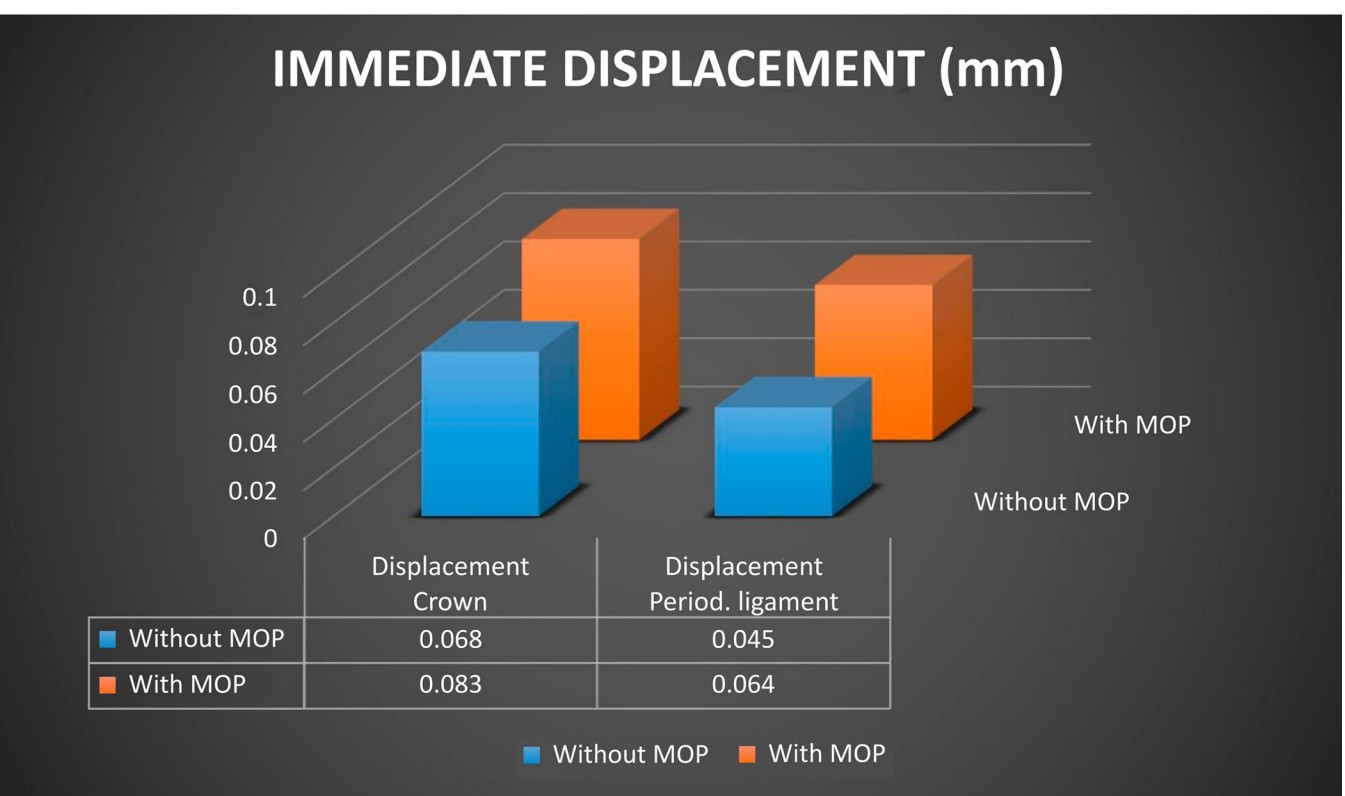

**Fig 15. Graph of immediate displacement (mm) on the dental crown and periodontal ligament without MOPs and with MOPs.**

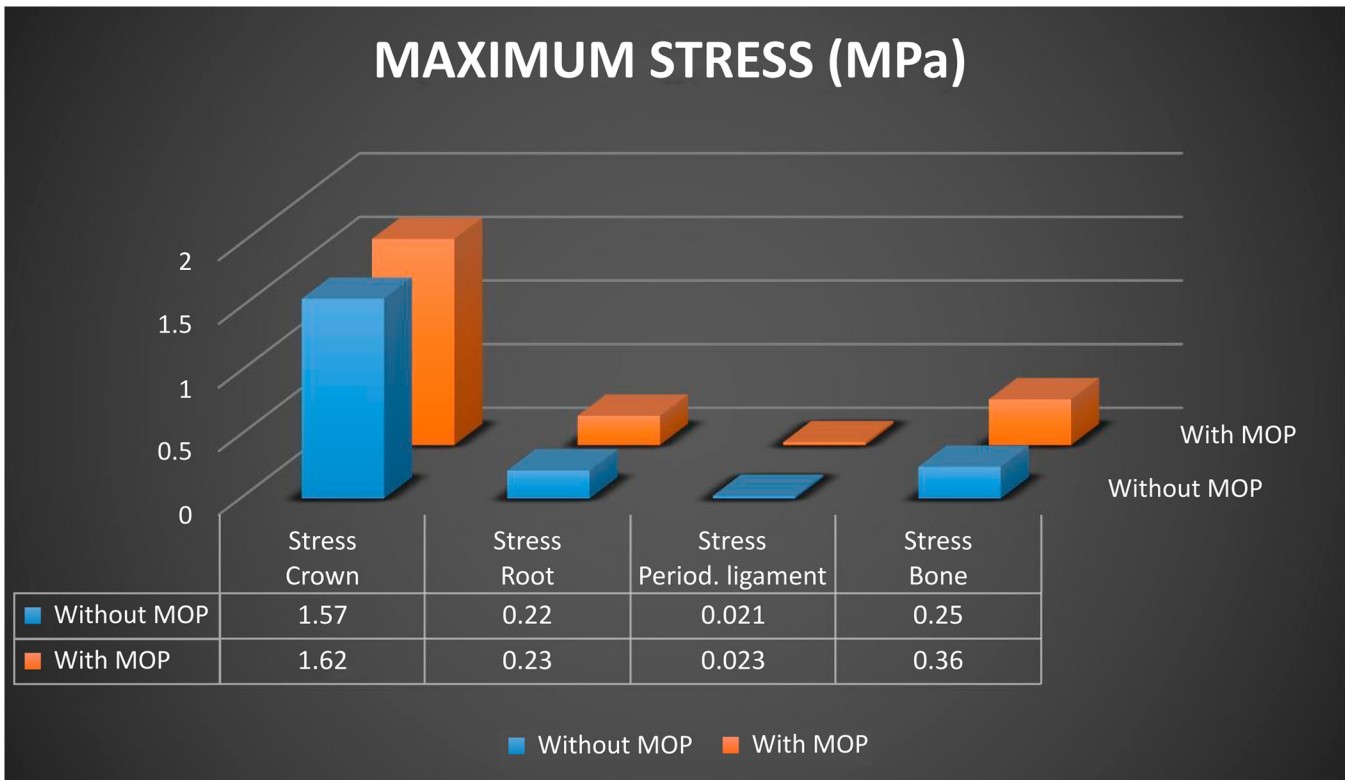

**Fig 16. Graph of maximum stress (MPa) on the dental crown, tooth root, periodontal ligament, and bone structure without MOPs and with MOPs.**

## Conclusions

The literature describing MOPs for accelerating orthodontic tooth movement is vast for the analysis of clinical and biological data, but not for elucidating the mechanical issues involved in the process of accelerating tooth movement using MOPs. The present study is the first to address and clearly describe the magnitude, distribution, and direction of the stress produced by orthodontic force on the tooth and supporting structures, also highlighting the need to understand the physiology of tooth movement so that, when translating results from a mathematical model into clinical practice, possible tissue damage can be minimized by controlling orthodontic force and biomechanics.

This study produced robust results by demonstrating, using FEM, that MOPs considerably increased tooth movement immediately after application of simulated orthodontic force, both on the dental crown and periodontal ligament, and also promoted larger stress distribution along the bone structure, with remarkable stress absorption regions being identified within the MOPs.

## Supporting information

**S1 File.**
(PDF)

**S2 File.**
(PDF)

**S3 File.**
(PDF)

**S4 File.**
(PDF)

**S5 File.**
(PDF)

**S6 File.**
(PDF)

**S7 File.**
(PDF)

**S8 File.**
(PDF)

**S1 Fig.**
(PDF)

**S2 Fig.**
(PDF)

**S3 Fig.**
(PDF)

**S4 Fig.**
(PDF)

**S5 Fig.**
(PDF)

**S6 Fig.**
(PDF)

## Acknowledgments

Our thanks to Dr. Daniel Ramos for his collaboration at the beginning of the study. We also thank Dr. Marcos Chevarria for his help with the scanning and capture of the Propel power tip, which were crucial for this study. We thank Bruno Passarela, technician at the Center for Diagnosis and Imaging (CDI), for image acquisition in STL files, which were essential for the fabrication of the virtual models.

## Author Contributions

**Conceptualization:** João Ricardo Cancian Lagomarcino Gomes, Ivana Ardenghi Vargas.

**Data curation:** Antônio Flávio Aires Rodrigues.

**Formal analysis:** Antônio Flávio Aires Rodrigues.

**Methodology:** Luiz Carlos Gertz, Pedro Antonio González Hernandez.

**Project administration:** Luiz Carlos Gertz, Pedro Antonio González Hernandez.

**Software:** Antônio Flávio Aires Rodrigues.

**Supervision:** Pedro Antonio González Hernandez.

**Validation:** Ivana Ardenghi Vargas, Ahmet Ozkomur, Pedro Antonio González Hernandez.

**Visualization:** João Ricardo Cancian Lagomarcino Gomes, Antônio Flávio Aires Rodrigues.

**Writing – original draft:** João Ricardo Cancian Lagomarcino Gomes.

**Writing – review & editing:** Ivana Ardenghi Vargas, Maria Perpétua Freitas, Sergio Augusto Quevedo Miguens, Jr.

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
