## [Decision Letter · Decision Letter 0]

6 May 2024

PONE-D-24-08264Micro-osteoperforation for enhancement of orthodontic movement: a mechanical analysis using the finite element methodPLOS ONE

Dear Dr. Gomes,

Thank you for submitting your manuscript to PLOS ONE. After careful consideration, we feel that it has merit but does not fully meet PLOS ONE’s publication criteria as it currently stands. Therefore, we invite you to submit a revised version of the manuscript that addresses the points raised during the review process.

We look forward to receiving your revised manuscript.

Kind regards,

Gaetano Isola, Ph.D.

Academic Editor

PLOS ONE

Additional Editor Comments:

Major revisions are necessary before nay further assessment of the manuscript.

Reviewers' comments:

Reviewer's Responses to Questions

**Comments to the Author**

1. Is the manuscript technically sound, and do the data support the conclusions?

Reviewer #1: Yes

Reviewer #2: Yes

2. Has the statistical analysis been performed appropriately and rigorously? 

Reviewer #1: Yes

Reviewer #2: Yes

3. Have the authors made all data underlying the findings in their manuscript fully available?

Reviewer #1: Yes

Reviewer #2: Yes

4. Is the manuscript presented in an intelligible fashion and written in standard English?

Reviewer #1: Yes

Reviewer #2: Yes

5. Review Comments to the Author

Reviewer #1: 1. On the perforation location in relation to the periodontal membrane: For sections 105-111, consider adding more details about the distance between the perforation location and the periodontal membrane, along with the spacing of the perforations. These factors are crucial in understanding how they affect tooth movement and stress on the periodontal membrane.

2. Observations regarding the periodontal membrane: In sections 190-192, observing the displacement of the periodontal membrane is not necessary because the data has limited significance. Instead, the focus should be on changes in hydrostatic pressure within the periodontal membrane, specifically monitoring the pressure values and rate of increase.

If the stress on the periodontal membrane remains within a reasonable range, it can promote tooth movement without causing excessive resorption. However, it's essential to consider that if the applied force is too high, it might lead to excessive stress, increasing the risk of damage or resorption.

Therefore, the analysis in this section should concentrate on understanding changes in hydrostatic pressure to ensure that stress stays within a range that supports safe tooth movement without causing damage or absorption in the periodontal membrane.

3. On the stress distribution in teeth: In sections 197-202, concentrate on the stress distribution in the tooth roots instead of the crowns. Analyzing stress growth and values in the roots is more meaningful to assess the risk of root damage.

4. On stress evaluation in the alveolar bone: In sections 207-210, use the distribution of minimum principal stress to assess the stress condition in the alveolar bone instead of relying solely on equivalent stress. Additionally, ensure that the stress distribution in the results is not exaggerated to avoid misleading readers.

5. On the quality of the images: Address the issue of the small size of numerical displays in the images. Remake high-definition images and ensure that the values are easily readable.

In summary, the surgical method discussed in the paper is innovative, but the data processing could be improved. Adding a discussion on stress distribution in the periodontal membrane and optimizing how experimental data is presented will increase the scientific accuracy and credibility of the paper.

Reviewer #2: What are the findings regarding the immediate tooth movement and stress distribution after the application of simulated orthodontic force with and without micro-osteoperforations?

How did the finite element method (FEM) demonstrate the impact of micro-osteoperforations on tooth movement and stress distribution on the dental crown, periodontal ligament, and bone structure?

What are the limitations of the study in terms of cellular and tissue response and how does the study contribute to filling a gap in the previously published clinical trials?

What are the potential implications of the study's findings on orthodontic treatment outcomes and the use of micro-osteoperforations in clinical practice?

6. PLOS authors have the option to publish the peer review history of their article (what does this mean?). If published, this will include your full peer review and any attached files.

Reviewer #1: **Yes: **wei wang

Reviewer #2: No

---

## [Author Response · Author response to Decision Letter 0]

19 Jul 2024

Gaetano Isola

Academic Editor

PLOS ONE

RE: PONE-D-24-08264, entitled "Micro-osteoperforation for enhancement of orthodontic movement: a mechanical analysis using the finite element method"

Dear Dr. Gaetano Isola

Thank you for your email dated May 6, 2024, and once again for the careful review of our manuscript, which we have amended following the reviewer’s suggestions. A marked-up copy of the revised manuscript (with track changes) as well as an unmarked copy of the manuscript (without track changes) have been uploaded to the submission system. Also, please find below an itemized point-by-point response to the reviewers’ comments.

We look forward to hearing from you about the status of our manuscript, which we hope is now acceptable for publication in PLOS ONE. Please feel free to contact me if you require any additional information. 

Sincerely,

João Ricardo Cancian Gomes

jr.cancian@hotmail.com

Additional Editor Comments:

Major revisions are necessary before nay further assessment of the manuscript.

Comments to the Author

1. Is the manuscript technically sound, and do the data support the conclusions?

Reviewer #1: Yes

Reviewer #2: Yes

2. Has the statistical analysis been performed appropriately and rigorously?

Reviewer #1: Yes

Reviewer #2: Yes

3. Have the authors made all data underlying the findings in their manuscript fully available?

Reviewer #1: Yes

Reviewer #2: Yes

4. Is the manuscript presented in an intelligible fashion and written in standard English?

Reviewer #1: Yes

Reviewer #2: Yes

5. Review Comments to the Author

Reviewer #1: 1. On the perforation location in relation to the periodontal membrane: For sections 105-111, consider adding more details about the distance between the perforation location and the periodontal membrane, along with the spacing of the perforations. These factors are crucial in understanding how they affect tooth movement and stress on the periodontal membrane.

Response: On average, the distance between the perforation location and the periodontal ligament was 1 mm, and the spacing of the perforations was 4.5 mm. However, it is important to highlight that this surgical technique was designed to be performed manually, by orthodontists themselves, in an easier way, without incisions or flaps, where the orthodontist only uses a panoramic radiograph to evaluate divergence or proximity of the roots and bone quantity and quality. The technique does not require the use of a rigid surgical guide, produced by CT and digital planning, to obtain more accurate drilling. The technique suggests making 3 perforations; however, performing micro-osteoperforations (MOPs) may not be possible due to the convergence or proximity of the roots. Therefore, standardizing a minimum spacing of the perforations or a minimum distance between the perforation location and the periodontal ligament would not reflect the clinical reality of the technique.

2. Observations regarding the periodontal membrane: In sections 190-192, observing the displacement of the periodontal membrane is not necessary because the data has limited significance. Instead, the focus should be on changes in hydrostatic pressure within the periodontal membrane, specifically monitoring the pressure values and rate of increase.

If the stress on the periodontal membrane remains within a reasonable range, it can promote tooth movement without causing excessive resorption. However, it's essential to consider that if the applied force is too high, it might lead to excessive stress, increasing the risk of damage or resorption.

Therefore, the analysis in this section should concentrate on understanding changes in hydrostatic pressure to ensure that stress stays within a range that supports safe tooth movement without causing damage or absorption in the periodontal membrane.

Response: Regarding the increase in displacement, immediately after the application of simulated orthodontic force, both from the crown and the periodontal ligament, there is greater MOP efficiency in the use and transformation of orthodontic force as well as in the displacement of the periodontal ligament to accelerate orthodontic movement, which, when related to biological principles, would result in greater bone remodeling. 

The periodontal ligament was considered solid based on the methodology used in the study. According to Serpe et al. (2015), the model used for the periodontal ligament (modulus of elasticity and Poisson coefficient) does not characterize the periodontal ligament as fluid, and for this reason it is unfeasible to consider changes in hydrostatic pressure. It is worth mentioning the studies by McCormack et al. (2014 and 2017), which consider that periodontal ligament fibers play a very important role in load transfer between the tooth and alveolar bone and their inclusion makes a significant difference to both the magnitude and distribution of strains produced in the surrounding bone and should be considered in studies using the finite element method (FEM) to investigate the biomechanics of orthodontic tooth movement.

As for the magnitude of the force used to simulate the orthodontic distalization force (150 gf ~= 1.5 N), it has been very well described in the literature in numerous clinical trials, and it was not the objective of this study to discuss it or establish another degree of intensity to be used.

3. On the stress distribution in teeth: In sections 197-202, concentrate on the stress distribution in the tooth roots instead of the crowns. Analyzing stress growth and values in the roots is more meaningful to assess the risk of root damage.

Response: The results of the study showed no difference in stress intensity and distribution in the roots with or without MOPs. Due to the high-fidelity virtual model produced, stress was distributed on the bone structure through the periodontal ligament. Therefore, the study showed the conductive function of the periodontal ligament for stress distribution produced by simulated orthodontic force, without the presence of areas of high stress distribution in the root. Of note, given the methodology used in laboratory experimental studies, it was not possible to perform statistical calculations.

4. On stress evaluation in the alveolar bone: In sections 207-210, use the distribution of minimum principal stress to assess the stress condition in the alveolar bone instead of relying solely on equivalent stress. Additionally, ensure that the stress distribution in the results is not exaggerated to avoid misleading readers.

Response: As the data provided by minimum principal stress elucidate the areas of compression (with negative values) and traction (with positive values), this being a well-known process in the biomechanics of tooth movement, in the study methodology we chose to use the Von Misses stress criterion to obtain a clearer result regarding the magnitude of maximum stress, as well as the stress distribution resulting from the simulated orthodontic force on the tooth and supporting structures. Nevertheless, we would like to thank the Reviewer for the suggestion to incorporate minimum principal stress into the methodology of future studies on the same topic: FEM x MOP relationship.

5. On the quality of the images: Address the issue of the small size of numerical displays in the images. Remake high-definition images and ensure that the values are easily readable.

Response: The images were revised according to the journal’s requirements and, whenever necessary, they were remade to ensure that the values are easily readable.

6. In summary, the surgical method discussed in the paper is innovative, but the data processing could be improved. Adding a discussion on stress distribution in the periodontal membrane and optimizing how experimental data is presented will increase the scientific accuracy and credibility of the paper.

Response: Regarding stress distribution in the periodontal ligament, no difference was observed with or without MOPs, based on the same explanations provided in comment 3. Due to the high-fidelity virtual model produced, stress was distributed on the bone structure through the periodontal ligament. Therefore, the study showed the conductive function of the periodontal ligament for stress distribution produced by simulated orthodontic force, without stress accumulation on the tooth structure (dental crown, root, and periodontal ligament). 

Reviewer #2: 1. What are the findings regarding the immediate tooth movement and stress distribution after the application of simulated orthodontic force with and without micro-osteoperforations?

Response: The immediate tooth movement on the dental crown region after the application of simulated orthodontic force had a maximum value of 0.0686 mm without micro-osteoperforations (MOPs) and 0.0900 mm with MOPs, being 24% higher with MOPs, while stress magnitude and distribution showed no significant numerical and area differences, whether in the dental crown, root, or periodontal ligament, either with or without MOPs. However, in the bone structure, there was an increase in the area of stress distribution after the application of simulated orthodontic force on the bone structure, with important stress absorption regions being identified within the MOPs.

2. How did the finite element method (FEM) demonstrate the impact of micro-osteoperforations on tooth movement and stress distribution on the dental crown, periodontal ligament, and bone structure?

Response: The applicability of and results obtained from FEM in dentistry seek to support satisfactory operative procedures that can make clinical practice more up-to-date, safe, and efficient. In the present study, FEM allowed us to control for variables reported in previous clinical studies, yielding results that indicate the effectiveness of the MOP technique. These findings include the larger initial displacement of the dental crown, which was 24% higher in the experimental model. Stress distribution was similar on the dental crown, root, and periodontal ligament with or without MOPs. However, on the bone structure, we could identify in the mathematical model a larger area of stress distribution, with important stress absorption regions being identified within the MOPs, highlighting the conductive function of the periodontal ligament for stress distribution produced by simulated orthodontic force, without the presence of areas of significantly high stress distribution on the root or periodontal ligament.

3. What are the limitations of the study in terms of cellular and tissue response and how does the study contribute to filling a gap in the previously published clinical trials?

Response: This study aims to contribute to the scarce literature on the topic and the difficulty of conducting “in vivo” clinical studies due to the complexity of standardizing a sample as well as quantifying and qualifying the results. A major limitation of using FEM is the inability to reproduce cellular and biological responses using a mathematical model due to the specificities and complexity of oral and craniofacial structures. It is important to keep in mind that the method and the precision of its results also have tolerance limits, without calling into question their reliability. The literature describing MOPs for accelerating orthodontic tooth movement is vast for the analysis of clinical and biological data, but not for elucidating the mechanical issues involved in the process of accelerating tooth movement using MOPs. The present study is the first to address and clearly describe the magnitude, distribution, and direction of the stress produced by orthodontic force on the tooth and supporting structures, also highlighting the need to understand the physiology of tooth movement so that, when translating results from a mathematical model into clinical practice, possible tissue damage can be minimized by controlling orthodontic force and biomechanics.

4. What are the potential implications of the study's findings on orthodontic treatment outcomes and the use of micro-osteoperforations in clinical practice?

Response: MOPs demonstrated effectiveness in accelerating orthodontic tooth movement by initially increasing tooth displacement not only by the applied forces and cell activation and migration, as shown in clinical and biological studies, but also by the subtraction of bone structure, producing less dense bone tissue. Extrapolating to clinical practice, this will likely result in a joint contribution of mechanical and biological factors to accelerate bone remodeling and, consequently, tooth movement.

---

## [Decision Letter · Decision Letter 1]

30 Jul 2024

Micro-osteoperforation for enhancement of orthodontic movement: a mechanical analysis using the finite element method

PONE-D-24-08264R1

Dear Dr. Gomes,

We’re pleased to inform you that your manuscript has been judged scientifically suitable for publication and will be formally accepted for publication once it meets all outstanding technical requirements.

Kind regards,

Gaetano Isola, Ph.D.

Academic Editor

PLOS ONE

Additional Editor Comments (optional):

The authors have made all the changes requested by the reviewers. The manuscript can be accepted for publication.

Reviewers' comments:

Reviewer's Responses to Questions

**Comments to the Author**

1. If the authors have adequately addressed your comments raised in a previous round of review and you feel that this manuscript is now acceptable for publication, you may indicate that here to bypass the “Comments to the Author” section, enter your conflict of interest statement in the “Confidential to Editor” section, and submit your "Accept" recommendation.

Reviewer #1: All comments have been addressed

2. Is the manuscript technically sound, and do the data support the conclusions?

Reviewer #1: (No Response)

3. Has the statistical analysis been performed appropriately and rigorously? 

Reviewer #1: (No Response)

4. Have the authors made all data underlying the findings in their manuscript fully available?

Reviewer #1: (No Response)

5. Is the manuscript presented in an intelligible fashion and written in standard English?

Reviewer #1: (No Response)

6. Review Comments to the Author

Reviewer #1: (No Response)

7. PLOS authors have the option to publish the peer review history of their article (what does this mean?). If published, this will include your full peer review and any attached files.

Reviewer #1: No

---

## [Editor Report · Acceptance letter]

7 Aug 2024

PONE-D-24-08264R1 

PLOS ONE

Dear Dr. Gomes, 

I'm pleased to inform you that your manuscript has been deemed suitable for publication in PLOS ONE. Congratulations! Your manuscript is now being handed over to our production team.

Kind regards, 

on behalf of

Prof. Gaetano Isola 

Academic Editor

PLOS ONE